# FAMO: Fast Adaptive Multitask Optimization

[†]**Bo Liu**, [‡]**Yihao Feng**, [†,§]**Peter Stone**, [†]**Qiang Liu**
[†]The University of Texas at Austin, [‡]Salesforce AI Research, [§]Sony AI
{bliu, pstone, lqiang}@cs.utexas.edu, yihaof@salesforce.com

## Abstract

One of the grand enduring goals of AI is to create generalist agents that can learn multiple different tasks from diverse data via multitask learning (MTL). However, in practice, applying gradient descent (GD) on the average loss across all tasks may yield poor multitask performance due to severe under-optimization of certain tasks. Previous approaches that manipulate task gradients for a more balanced loss decrease require storing and computing all task gradients ($\mathcal{O}(k)$ space and time where $k$ is the number of tasks), limiting their use in large-scale scenarios. In this work, we introduce Fast Adaptive Multitask Optimization (FAMO), a dynamic weighting method that decreases task losses in a balanced way using $\mathcal{O}(1)$ space and time. We conduct an extensive set of experiments covering multi-task supervised and reinforcement learning problems. Our results indicate that FAMO achieves comparable or superior performance to state-of-the-art gradient manipulation techniques while offering significant improvements in space and computational efficiency. Code is available at `https://github.com/Cranial-XIX/FAMO`.

## 1 Introduction

Large models trained on diverse data have advanced both computer vision [20] and natural language processing [4], paving the way for generalist agents capable of multitask learning (MTL) [5]. Given the substantial size of these models, it is crucial to design MTL methods that are *effective* in terms of task performance and *efficient* in terms of space and time complexities for managing training costs and environmental impacts. This work explores such methods through the lens of optimization.

Perhaps the most intuitive way of solving an MTL problem is to optimize the average loss across all tasks. However, in practice, doing so can lead to models with poor multitask performance: a subset of tasks are *severely under-optimized*. A major reason behind such optimization failure is that a subset of tasks are under-optimized because the average gradient constantly results in small (or even negative) progress on these tasks (see details in Section 2).

To mitigate this problem, gradient manipulation methods [43, 25, 7, 24] compute a new update vector in place of the gradient to the average loss, such that all task losses decrease in a more balanced way. The new update vector is often determined by solving an additional optimization problem that involves all task gradients. While these approaches exhibit improved performance, they become computationally expensive when the number of tasks and the model size are large [41]. This is because they require computing and storing all task gradients at each iteration, thus demanding $\mathcal{O}(k)$ space and time complexities, not to mention the overhead introduced by solving the additional optimization problem. In contrast, the average gradient can be efficiently computed in $\mathcal{O}(1)$ space and time per iteration because one can first average the task losses and then take the gradient of the average loss.[1] To this end, we ask the following question:

---

[1]Here, we refer to the situation where a single data $x$ can be used to compute all task losses.

37th Conference on Neural Information Processing Systems (NeurIPS 2023).

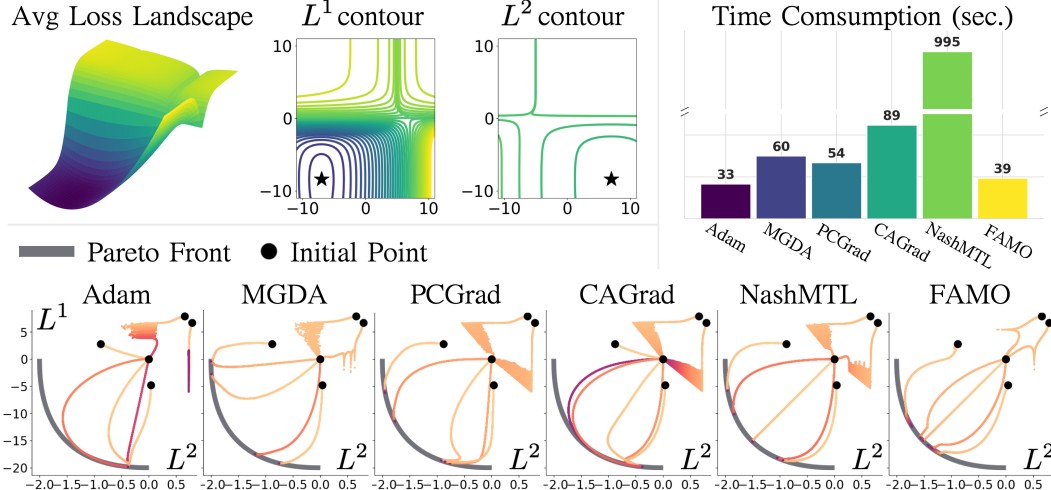

**Figure 1: Top left:** The loss landscape, and individual task losses of a toy 2-task learning problem ($\star$ represents the minimum of task losses). **Top right:** the runtime of different MTL methods for 50000 steps. **Bottom:** the loss trajectories of different MTL methods. ADAM fails in 1 out of 5 runs to reach the Pareto front due to CG. FAMO decreases task losses in a balanced way and is the only method matching the $\mathcal{O}(1)$ space/time complexity of ADAM. Experimental details and analysis are provided in Section 5.1.

$(Q)$ *Is it possible to design a multi-task learning optimizer that ensures a balanced reduction in losses across all tasks while utilizing $\mathcal{O}(1)$ space and time per iteration?*

In this work, we present Fast Adaptive Multitask Optimization (FAMO), a simple yet effective adaptive task weighting method to address the above question. On the one hand, FAMO is designed to ensure that all tasks are optimized with approximately similar progress. On the other hand, FAMO leverages the loss history to update the task weighting, hence bypassing the necessity of computing all task gradients. To summarize, our contributions are:

1. We introduce FAMO, an MTL optimizer that decreases task losses approximately at *equal rates* while using only $\mathcal{O}(1)$ space and time per iteration.

2. We demonstrate that FAMO performs comparably to or better than existing gradient manipulation methods on a wide range of standard MTL benchmarks, in terms of standard MTL metrics, while being significantly computationally cheaper.

## 2 Background

In this section, we provide the formal definition of multitask learning, then discuss its optimization challenge, and provide a brief overview of the gradient manipulation methods.

**Multitask Learning (MTL)**   MTL considers optimizing a *single* model with parameter $\theta \in \mathbb{R}^m$ that can perform $k \geq 2$ tasks well, where each task is associated with a loss function $\ell_i(\theta) : \mathbb{R}^m \to \mathbb{R}_{\geq 0}$.[2] Then, it is common to optimize the average loss across all tasks:

$$\min_{\theta \in \mathbb{R}^m} \left\{ \ell_0(\theta) := \frac{1}{k} \sum_{i=1}^{k} \ell_i(\theta) \right\}. \tag{1}$$

**Optimization Challenge**   Directly optimizing (1) can result in severe under-optimization of a subset of tasks. A major reason behind this optimization challenge is the "generalized" conflicting gradient phenomenon, which we explain in the following. At any time step $t$, assume one updates the model

---

[2]In this work, we assume $\forall\ i,\ \ell_i(\theta) \geq 0$, which is true for typical loss functions including mean square and cross-entropy losses. Note that one can always transform $\ell_i$ to be non-negative if a loss lower bound is known.

---

**Algorithm 1** Fast Adaptive Multitask Optimization (FAMO)

---

1: **Input**: Initial parameter $\theta_0$, task losses $\{\ell_i\}_{i=1}^{k}$ (ensure that $\ell_i \geq \epsilon > 0$, for instance, by $\ell_i \leftarrow \ell_i - \ell_i^* + \epsilon$, $\ell_i^* = \inf_\theta \ell_i(\theta)$), learning rate $\alpha$ and $\beta$, and decay $\gamma$ (= 0.001 by default).
2: $\xi_1 \leftarrow 0$.     **// initialize the task logits to all zeros**
3: **for** $t = 1 : T$ **do**
4:     Compute $z_t = \textbf{Softmax}(\xi_t)$, e.g.,

$$z_{i,t} = \frac{\exp(\xi_{i,t})}{\sum_{i=1}^{k} \exp(\xi_{i,t})}.$$

5:     Update the model parameters:

$$\theta_{t+1} = \theta_t - \alpha \sum_{i=1}^{k} \Big(c_t \frac{z_{i,t}}{\ell_{i,t}}\Big)\nabla\ell_{i,t}, \ \ \text{where} \ \ c_t = \Big(\sum_{i=1}^{k} \frac{z_{i,t}}{\ell_{i,t}}\Big)^{-1}.$$

6:     Update the logits for task weighting:

$$\xi_{t+1} = \xi_t - \beta\big(\delta_t + \gamma\xi_t\big) \ \ \text{where} \ \ \delta_t = \begin{bmatrix} \nabla^\top z_{1,t}(\xi_t) \\ \vdots \\ \nabla^\top z_{k,t}(\xi_t) \end{bmatrix}^\top \begin{bmatrix} \log\ell_{1,t} - \log\ell_{1,t+1} \\ \vdots \\ \log\ell_{k,t} - \log\ell_{k,t+1}. \end{bmatrix}.$$

7: **end for**

---

parameter using a gradient descent style iterative update: $\theta_{t+1} = \theta_t - \alpha d_t$ where $\alpha$ is the step size and $d_t$ is the update at time $t$. Then, we say that conflicting gradients (CG) [24, 43] happens if

$$\exists i, \ \ \ell_i(\theta_{t+1}) - \ell_i(\theta_t) \approx -\alpha\nabla\ell_i(\theta_t)^\top d_t > 0.$$

In other words, certain task's loss is increasing. CG often occurs during optimization and is not inherently detrimental. However, it becomes undesirable when a subset of tasks persistently undergoes under-optimization due to CG. In a more general sense, it is not desirable if a subset of tasks has much slower learning progress compared to the rest of the tasks (even if all task losses are decreasing). This very phenomenon, which we call the "generalized" conflicting gradient, has spurred previous research to mitigate it at each optimization stage [43].

**Gradient Manipulation Methods**   Gradient manipulation methods aim to decrease all task losses in a more balanced way by finding a new update $d_t$ at each step. $d_t$ is usually a convex combination of task gradients, and therefore the name gradient manipulation (denote $\nabla\ell_{i,t} = \nabla_\theta\ell_i(\theta_t)$ for short):

$$d_t = \begin{bmatrix} \nabla\ell_{1,t}^\top \\ \vdots \\ \nabla\ell_{k,t}^\top \end{bmatrix}^\top w_t, \quad \text{where} \ \ w_t = \begin{bmatrix} w_{1,t} \\ \vdots \\ w_{k,t} \end{bmatrix} = f\big(\nabla\ell_{1,t}, \ldots, \nabla\ell_{k,t}\big) \in \mathbb{S}_k. \tag{2}$$

Here, $\mathbb{S}_k = \{w \in \mathbb{R}_{\geq 0}^k \mid w^\top \mathbf{1} = 1\}$ is the probabilistic simplex, and $w_t$ is the task weighting across all tasks. Please refer to Appendix A for details of five state-of-the-art gradient manipulation methods (MGDA, PCGRAD, CAGRAD, IMTL-G, NASHMTL) and their corresponding $f$. Note that existing gradient manipulation methods require computing and storing $k$ task gradients before applying $f$ to compute $d_t$, which often involves solving an additional optimization problem. As a result, we say these methods require at least $\mathcal{O}(k)$ space and time complexity, which makes them slow and memory inefficient when $k$ and model size $m$ are large.

## 3   Fast Adaptive Multitask Optimization (FAMO)

In this section, we introduce FAMO that addresses question $Q$, which involves two main ideas:

1. At each step, decrease all task losses at *an equal rate* as much as possible (Section 3.1).

2. Amortize the computation in 1. over time (Section 3.2).

### 3.1 Balanced Rate of Loss Improvement

At time $t$, assume we perform the update $\theta_{t+1} = \theta_t - \alpha d_t$, we define the rate of improvement for task $i$ as

$$r_i(\alpha, d_t) = \frac{\ell_{i,t} - \ell_{i,t+1}}{\ell_{i,t}}.^3 \tag{3}$$

FAMO then seeks an update $d_t$ that results in the largest *worst-case improvement rate* across all tasks ($\frac{1}{2}\|d_t\|$ is subtracted to prevent an under-specified optimization problem where the objective can be infinitely large):

$$\max_{d_t \in \mathbb{R}^m} \min_{i \in [k]} \frac{1}{\alpha} r_i(\alpha, d_t) - \frac{1}{2}\|d_t\|^2. \tag{4}$$

When the step size $\alpha$ is small, using Taylor approximation, the problem (4) can be approximated by

$$\max_{d_t \in \mathbb{R}^m} \min_{i \in [K]} \frac{\nabla \ell_{i,t}^\top d_t}{\ell_{i,t}} - \frac{1}{2}\|d_t\|^2 = \left(\nabla \log \ell_{i,t}\right)^\top d_t - \frac{1}{2}\|d_t\|^2. \tag{5}$$

Instead of solving the primal problem in (5) where $d \in \mathbb{R}^m$ ($m$ can be millions if $\theta$ is the parameter of a neural network), we consider its dual problem:

**Proposition 3.1.** *The dual objective of* (5) *is*

$$z_t^* \in \arg\min_{z \in \mathbb{S}_k} \frac{1}{2}\|J_t z\|^2, \quad where \quad J_t = \begin{bmatrix} \nabla \log \ell_{1,t}^\top \\ \vdots \\ \nabla \log \ell_{k,t}^\top \end{bmatrix}, \tag{6}$$

*where $z_t^* = [z_{t,i}^*]$ is the optimal combination weights of the gradients, and the optimal update direction is $d_t^* = J_t z_t^*$.*

*Proof.*

$$\max_{d \in \mathbb{R}^m} \min_{i \in [k]} \left(\nabla \log \ell_{i,t}\right)^\top d - \frac{1}{2}\|d\|^2$$

$$= \max_{d \in \mathbb{R}^m} \min_{z \in \mathbb{S}_k} \left(\sum_{i=1}^k z_i \nabla \log \ell_{i,t}\right)^\top d - \frac{1}{2}\|d\|^2$$

$$= \min_{z \in \mathbb{S}_k} \max_{d \in \mathbb{R}^m} \left(\sum_{i=1}^k z_i \nabla \log \ell_{i,t}\right)^\top d - \frac{1}{2}\|d\|^2 \qquad \text{(strong duality)}$$

Write $g(d, z) = \left(\sum_{i=1}^k z_i \nabla \log \ell_{i,t}\right)^\top d - \frac{1}{2}\|d\|^2$, then by setting

$$\frac{\partial g}{\partial d} = 0 \quad \Longrightarrow \quad d^* = \sum_{i=1}^k z_i \nabla \log \ell_{i,t}.$$

Plugging in $d^*$ back, we have

$$\max_{d \in \mathbb{R}^m} \min_{i \in [k]} \left(\nabla \log \ell_{i,t}\right)^\top d - \frac{1}{2}\|d\|^2 = \min_{z \in \mathbb{S}_k} \frac{1}{2}\left\|\sum_{i=1}^k z_i \nabla \log \ell_{i,t}\right\|^2 = \min_{z \in \mathbb{S}_k} \frac{1}{2}\|J_t z\|^2.$$

At the optimum, we have $d_t^* = J_t z_t^*$. $\qquad\qquad\square$

The dual problem in (6) can be viewed as optimizing the log objective of the multiple gradient descent algorithm (MGDA) [9, 35]. Similar to MGDA, (6) only involves a decision variable of dimension $k \ll m$. Furthermore, if the optimal combination weights $z_t^*$ is an interior point of $\mathbb{S}_k$, then the improvement rates $r_i(\alpha, d_t^*)$ of the different tasks $i$ equal, as we show in the following result.

---

[3]To avoid division by zero, in practice we add a small constant (e.g., $1e-8$) to all losses. For the ease of notation (e.g., $\ell_i(\cdot) \leftarrow \ell_i(\cdot) + 1e-8$), we omit it throughout the paper.

**Proposition 3.2.** *Assume* $\{\ell_i\}_{i=1}^k$ *are smooth and the optimal weights* $z_t^*$ *in* (6) *is an interior point of* $\mathbb{S}_k$, *then*

$$\forall\, i \neq j \in [k], \qquad r_i^*(d_t^*) = r_j^*(d_t^*),$$

*where* $r_i^*(d_t^*) = \lim_{\alpha \to 0} \frac{1}{\alpha} r_i(\alpha, d_t^*)$.

*Proof.* Consider the Lagrangian form of (6)

$$\mathcal{L}(z, \lambda, \mu) = \frac{1}{2}\left\|\sum_{i=1}^k z_i \nabla \log \ell_{i,t}\right\|^2 + \lambda\Big(\sum_{i=1}^k z_i - 1\Big) - \sum_{i=1}^k \mu_i z_i, \ \text{ where } \forall i, \mu_i \geq 0. \tag{7}$$

When $z^*$ reaches the optimum, we have $\partial \mathcal{L}(z, \lambda, \mu)/\partial z = 0$, recall that $d_t^* = J_t z_t^*$, then

$$J_t^\top J_t z^* = -\mu - \lambda, \quad \text{where} \quad J_t = \begin{bmatrix} \nabla \log \ell_{1,t}^\top \\ \vdots \\ \nabla \log \ell_{k,t}^\top \end{bmatrix} \implies J_t^\top d_t^* = -(\mu + \lambda).$$

When $z_t^*$ is an interior point of $\mathbb{S}_k$, we know that $\mu = 0$. Hence $J_t^\top d_t^* = -\lambda$. This means,

$$\forall i \neq j, \qquad \lim_{\alpha \to 0} \frac{1}{\alpha} r_i(\alpha, d_t^*) = \nabla \log \ell_{i,t}^\top d_t^* = \nabla \log \ell_{j,t}^\top d_t^* = \lim_{\alpha \to 0} \frac{1}{\alpha} r_j(\alpha, d_t^*).$$

$\square$

## 3.2 Fast Approximation by Amortizing over Time

Instead of fully solving (6) at each optimization step, FAMO performs a single-step gradient descent on $z$, which amortizes the computation over the optimization trajectory:

$$z_{t+1} = z_t - \alpha_z \tilde{\delta}, \quad \text{where } \tilde{\delta} = \nabla_z \frac{1}{2}\left\|\sum_{i=1}^k z_{i,t} \nabla \log \ell_{i,t}\right\|^2 = J_t^\top J_t z_t. \tag{8}$$

But then, note that

$$\frac{1}{\alpha}\begin{bmatrix} \log \ell_{1,t} - \log \ell_{1,t+1} \\ \vdots \\ \log \ell_{k,t} - \log \ell_{k,t+1} \end{bmatrix} \approx J_t^\top d_t = J_t^\top J_t z_t, \tag{9}$$

so we can use the change in log losses to approximate the gradient.

In practice, to ensure that $z$ always stays in $\mathbb{S}_k$, we re-parameterize $z$ by $\xi$ and let $z_t = \textbf{Softmax}(\xi_t)$, where $\xi_t \in \mathbb{R}^K$ are the unconstrained softmax logits. Consequently, we have the following approximate update on $\xi$ from (8):

$$\xi_{t+1} = \xi_t - \beta\delta, \quad \text{where } \delta = \begin{bmatrix} \nabla^\top z_{1,t}(\xi) \\ \vdots \\ \nabla^\top z_{k,t}(\xi) \end{bmatrix}^\top \begin{bmatrix} \log \ell_{1,t} - \log \ell_{1,t+1} \\ \vdots \\ \log \ell_{k,t} - \log \ell_{k,t+1} \end{bmatrix}. \tag{10}$$

**Remark:** While it is possible to perform gradient descent on $z$ for other gradient manipulation methods in principle, we will demonstrate in Appendix B that not all such updates can be easily approximated using the change in losses.

## 3.3 Practical Implementation

To facilitate practical implementation, we present two modifications to the update in (10).

**Re-normalization** The suggested update above is a convex combination of the gradients of the log loss, e.g.,

$$d^* = \sum_{i=1}^k z_{i,t} \nabla \log \ell_{i,t} = \sum_{i=1}^k \Big(\frac{z_{i,t}}{\ell_{i,t}}\Big) \nabla \ell_{i,t}.$$

When $\ell_{i,t}$ is small, the multiplicative coefficient $\frac{z_{i,t}}{\ell_{i,t}}$ can be quite large and result in unstable optimization. Therefore, we propose to multiply $d^*$ by a constant $c_t$, such that $c_t d^*$ can be written as a convex combination of the task gradients just as in other gradient manipulation algorithms (see (2) and we provide the corresponding definition of $w$ in the following):

$$c_t = \Big(\sum_{i=1}^k \frac{z_{i,t}}{\ell_{i,t}}\Big)^{-1} \quad \text{and} \quad d_t = c_t d^* = \sum_{i=1}^k w_i \nabla \ell_{i,t}, \quad \text{where } w_i = c_t \frac{z_{i,t}}{\ell_{i,t}}. \tag{11}$$

**Regularization**    As we are amortizing the computation over time and the loss objective $\{\ell_i(\cdot)\}$s are changing dynamically, it makes sense to focus more on the recent updates of $\xi$ [46]. To this end, we put a decay term on $w$ such that the resulting $\xi_t$ is an exponential moving average of its gradient updates:

$$\xi_{t+1} = \xi_t - \beta(\delta_t + \gamma\xi_t) = -\beta(\delta_t + (1-\beta\gamma)\delta_{t-1} + (1-\beta\gamma)^2\delta_{t-2} + \dots). \tag{12}$$

We provide the complete FAMO algorithm in Algorithm 1 and its pseudocode in Appendix C.

## 3.4    The Continuous Limit of FAMO

One way to characterize FAMO's behavior is to understand the stationary points of the continuous-time limit of FAMO (i.e. when step sizes $(\alpha, \beta)$ shrink to zero). From Algorithm 1, one can derive the following non-autonomous dynamical system (assuming $\{\ell_i\}$ are all smooth):

$$\begin{bmatrix} \dot{\theta} \\ \dot{\xi} \end{bmatrix} = -c_t \begin{bmatrix} J_t z_t \\ A_t J_t^\top J_t z_t + \frac{\gamma}{c_t}\xi_t \end{bmatrix}, \quad \text{where } A_t = \begin{bmatrix} \nabla^\top z_{1,t}(\xi_t) \\ \vdots \\ \nabla^\top z_{k,t}(\xi_t) \end{bmatrix}. \tag{13}$$

(13) reaches its stationary points (or fixed points) when (note that $c_t > 0$)

$$\begin{bmatrix} \dot{\theta} \\ \dot{\xi} \end{bmatrix} = 0 \implies J_t z_t = 0 \text{ and } \xi_t = 0 \implies \sum_{i=1}^k \nabla \log \ell_{i,t} = 0. \tag{14}$$

Therefore, the minimum points of $\sum_{i=1}^k \log \ell_i(\theta)$ are all stationary points of (13).

# 4    Related Work

In this section, we summarize existing methods that tackle learning challenges in multitask learning (MTL). The general idea of most existing works is to encourage positive knowledge transfer by sharing parameters while decreasing any potential negative knowledge transfer (a.k.a, interference) during learning. There are three major ways of doing so: task grouping, designing network architectures specifically for MTL, and designing multitask optimization methods.

**Task Grouping**    Task grouping refers to grouping $K$ tasks into $N < K$ clusters and learning $N$ models for each cluster. The key is estimating the amount of positive knowledge transfer incurred by grouping certain tasks together and then identifying which tasks should be grouped [39, 45, 38, 36, 11].

**Multitask Architecture**    Novel neural architectures for MTL include *hard-parameter-sharing* methods, which decompose a neural network into task-specific modules and a shared feature extractor using manually designed heuristics [21, 29, 2], and *soft-parameter-sharing* methods, which learn which parameters to share [30, 34, 12, 27]. Recent studies extend neural architecture search for MTL by learning where to branch a network to have task-specific modules [14, 3].

**Multitask Optimization**    The most relevant approach to our method is MTL optimization via task balancing. These methods dynamically re-weight all task losses to mitigate the conflicting gradient issue [40, 43]. The simplest form of gradient manipulation is to re-weight the task losses based on manually designed criteria [6, 13, 18], but these methods are often heuristic and lack theoretical support. Gradient manipulation methods [35, 43, 25, 7, 16, 24, 32, 26, 47] propose to form a new update vector at each optimization by linearly combining task gradients. The local improvements across all tasks using the new update can often be explicitly analyzed, making these methods better understood in terms of convergence. However, it has been observed that gradient manipulation methods are often slow in practice, which may outweigh their performance benefits [22]. By contrast, FAMO is designed to match the performance of these methods while remaining efficient in terms of memory and computation. Another recent work proposes to sample random task weights at each optimization step for MTL [23], which is also computationally efficient. However, we will demonstrate empirically that FAMO performs better than this method.

# 5 Empirical Results

We conduct experiments to answer the following question:

*How does* FAMO *perform in terms of space/time complexities and standard MTL metrics against prior MTL optimizers on standard benchmarks (e.g., supervised and reinforcement MTL problems)?*

In the following, we first use a toy 2-task problem to demonstrate how FAMO mitigates CG while being efficient. Then we show that FAMO performs comparably or even better than state-of-the-art gradient manipulation methods on standard multitask supervised and reinforcement learning benchmarks. In addition, FAMO requires significantly lower computation time when $K$ is large compared to other methods. Lastly, we conduct an ablation study on how robust FAMO is to $\gamma$. Each subsection first details the experimental setup and then analyzes the results.

## 5.1 A Toy 2-Task Example

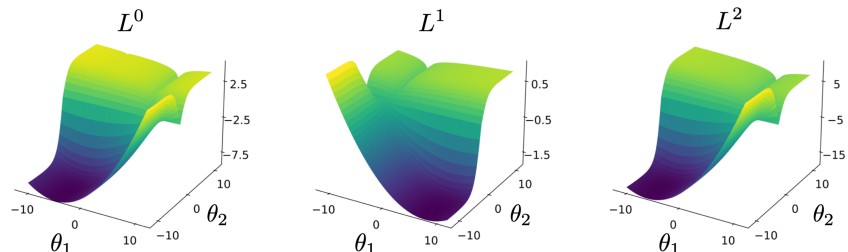

**Figure 2:** The average loss $L^0$ and the two task losses $L^1$ and $L^2$ for the toy example.

To better understand the optimization trajectory of FAMO, we adopt the same 2D multitask optimization problem from NASHMTL [32] to visualize how FAMO balances different loss objectives. The model parameter $\theta = (\theta_1, \theta_2) \in \mathbb{R}^2$. The two tasks' objectives and their surface plots are provided in Appendix D and Figure 2. We compare FAMO against ADAM [19], MGDA [35], PCGRAD [43], CAGRAD [24], and NASHMTL [32]. We then pick 5 initial points $\theta_{\text{init}} \in \{(-8.5, 7.5), (-8.5, 5), (0, 0), (9, 9), (10, -8)\}$ and plot the corresponding optimization trajectories with different methods in Figure 1. Note that the toy example is constructed such that naively applying ADAM on the average loss can cause the failure of optimization for task 1.

**Findings:** From Figure 1, we observe that FAMO, like all other gradient manipulation methods, mitigates the CG and reaches the Pareto front for all five runs. In the meantime, FAMO performs similarly to NASHMTL and achieves a balanced loss decrease even when the two task losses are improperly scaled. Finally, as shown in the top-right of the plot, FAMO behaves similarly to ADAM in terms of the training time, which is 25× faster than NASHMTL.

## 5.2 MTL Performance

**Multitask Supervised Learning.** We consider four supervised benchmarks commonly used in prior MTL research [24, 27, 32, 33]: NYU-v2 [31] (3 tasks), CityScapes [8] (2 tasks), QM-9 [1] (11 tasks), and CelebA [28] (40 tasks). Specifically, NYU-v2 is an indoor scene dataset consisting of 1449 RGBD images and dense per-pixel labeling with 13 classes. The learning objectives include image segmentation, depth prediction, and surface normal prediction based on any scene image. CityScapes dataset is similar to NYU-v2 but contains 5000 street-view RGBD images with per-pixel annotations. QM-9 dataset is a widely used benchmark in graph neural network learning. It consists of >130K molecules represented as graphs annotated with node and edge features. We follow the same experimental setting used in NASHMTL [32], where the learning objective is to predict 11 properties of molecules. We use 110K molecules from the QM9 example in PyTorch Geometric [10], 10K molecules for validation, and the rest of 10K molecules for testing. The characteristic of this dataset is that the 11 properties are at different scales, posing a challenge for task balancing in MTL. Lastly, CelebA dataset contains 200K face images of 10K different celebrities, and each face image is provided with 40 facial binary attributes. Therefore, CelebA can be viewed as a 40-task MTL problem. Different from NYU-v2, CityScapes, and QM-9, the number of tasks ($K$) in CelebA is much larger, hence posing a challenge to learning efficiency.

| Method | Segmentation | | Depth | | Surface Normal | | | | | MR ↓ | $\Delta m$% ↓ |
|---|---|---|---|---|---|---|---|---|---|---|---|
| | | | | | Angle Dist ↓ | | Within $t°$ ↑ | | | | |
| | mIoU ↑ | Pix Acc ↑ | Abs Err ↓ | Rel Err ↓ | Mean | Median | 11.25 | 22.5 | 30 | | |
| STL | 38.30 | 63.76 | 0.6754 | 0.2780 | 25.01 | 19.21 | 30.14 | 57.20 | 69.15 | | |
| LS | 39.29 | 65.33 | 0.5493 | 0.2263 | 28.15 | 23.96 | 22.09 | 47.50 | 61.08 | 8.89 | 5.59 |
| SI | 38.45 | 64.27 | 0.5354 | 0.2201 | 27.60 | 23.37 | 22.53 | 48.57 | 62.32 | 7.89 | 4.39 |
| RLW | 37.17 | 63.77 | 0.5759 | 0.2410 | 28.27 | 24.18 | 22.26 | 47.05 | 60.62 | 11.22 | 7.78 |
| DWA | 39.11 | 65.31 | 0.5510 | 0.2285 | 27.61 | 23.18 | 24.17 | 50.18 | 62.39 | 7.67 | 3.57 |
| UW | 36.87 | 63.17 | 0.5446 | 0.2260 | 27.04 | 22.61 | 23.54 | 49.05 | 63.65 | 7.44 | 4.05 |
| MGDA | 30.47 | 59.90 | 0.6070 | 0.2555 | 24.88 | 19.45 | 29.18 | 56.88 | 69.36 | 6.00 | 1.38 |
| PCGRAD | 38.06 | 64.64 | 0.5550 | 0.2325 | 27.41 | 22.80 | 23.86 | 49.83 | 63.14 | 8.00 | 3.97 |
| GRADDROP | 39.39 | 65.12 | 0.5455 | 0.2279 | 27.48 | 22.96 | 23.38 | 49.44 | 62.87 | 7.00 | 3.58 |
| CAGRAD | 39.79 | 65.49 | 0.5486 | 0.2250 | 26.31 | 21.58 | 25.61 | 52.36 | 65.58 | 4.56 | 0.20 |
| IMTL-G | 39.35 | 65.60 | 0.5426 | 0.2256 | 26.02 | 21.19 | 26.20 | 53.13 | 66.24 | 3.78 | -0.76 |
| NASHMTL | **40.13** | **65.93** | **0.5261** | **0.2171** | 25.26 | 20.08 | 28.40 | 55.47 | 68.15 | **2.11** | -4.04 |
| FAMO | 38.88 | 64.90 | 0.5474 | 0.2194 | 25.06 | 19.57 | **29.21** | 56.61 | 68.98 | 3.44 | **-4.10** |

**Table 1:** Results on NYU-v2 dataset (3 tasks). Each experiment is repeated over 3 random seeds and the mean is reported. The best average result is marked in bold. **MR** and $\Delta m$% are the main metrics for MTL performance.

| Method | $\mu$ | $\alpha$ | $\epsilon_{\text{HOMO}}$ | $\epsilon_{\text{LUMO}}$ | $\langle R^2 \rangle$ | ZPVE | $U_0$ | $U$ | $H$ | $G$ | $c_v$ | MR ↓ | $\Delta m$% ↓ |
|---|---|---|---|---|---|---|---|---|---|---|---|---|---|
| | | | | | | MAE ↓ | | | | | | | |
| STL | 0.07 | 0.18 | 60.6 | 53.9 | 0.50 | 4.53 | 58.8 | 64.2 | 63.8 | 66.2 | 0.07 | | |
| LS | 0.11 | 0.33 | **73.6** | 89.7 | 5.20 | 14.06 | 143.4 | 144.2 | 144.6 | 140.3 | 0.13 | 6.45 | 177.6 |
| SI | 0.31 | 0.35 | 149.8 | 135.7 | **1.00** | **4.51** | **55.3** | **55.8** | **55.8** | **55.3** | 0.11 | 3.55 | 77.8 |
| RLW | 0.11 | 0.34 | 76.9 | 92.8 | 5.87 | 15.47 | 156.3 | 157.1 | 157.6 | 153.0 | 0.14 | 8.00 | 203.8 |
| DWA | 0.11 | 0.33 | 74.1 | 90.6 | 5.09 | 13.99 | 142.3 | 143.0 | 143.4 | 139.3 | 0.13 | 6.27 | 175.3 |
| UW | 0.39 | 0.43 | 166.2 | 155.8 | 1.07 | 4.99 | 66.4 | 66.8 | 66.8 | 66.2 | 0.12 | 4.91 | 108.0 |
| MGDA | 0.22 | 0.37 | 126.8 | 104.6 | 3.23 | 5.69 | 88.4 | 89.4 | 89.3 | 88.0 | 0.12 | 5.91 | 120.5 |
| PCGRAD | 0.11 | 0.29 | 75.9 | 88.3 | 3.94 | 9.15 | 116.4 | 116.8 | 117.2 | 114.5 | 0.11 | 4.73 | 125.7 |
| CAGRAD | 0.12 | 0.32 | 83.5 | 94.8 | 3.22 | 6.93 | 114.0 | 114.3 | 114.5 | 112.3 | 0.12 | 5.45 | 112.8 |
| IMTL-G | 0.14 | 0.29 | 98.3 | 93.9 | 1.75 | 5.70 | 101.4 | 102.4 | 102.0 | 100.1 | 0.10 | 4.36 | 77.2 |
| NASHMTL | **0.10** | **0.25** | 82.9 | **81.9** | 2.43 | 5.38 | 74.5 | 75.0 | 75.1 | 74.2 | **0.09** | 2.09 | 62.0 |
| FAMO | 0.15 | 0.30 | 94.0 | 95.2 | 1.63 | 4.95 | 70.82 | 71.2 | 71.2 | 70.3 | 0.10 | 3.27 | **58.5** |

**Table 2:** Results on QM-9 dataset (11 tasks). Each experiment is repeated over 3 random seeds and the mean is reported. The best average result is marked in bold. **MR** and $\Delta m$% are the main metrics for MTL performance.

We compare FAMO against 11 MTL optimization methods and a single-task learning baseline: **(1)** Single task learning (STL), training an independent model ($\theta$ for each task; **(2)** Linear scalarization (LS) baseline that minimizes $L^0$; **(3)** Scale-invariant (SI) baseline that minimizes $\sum_k \log L^k(\theta)$, as SI is invariant to any scalar multiplication of task losses; **(4)** Dynamic Weight Average (DWA) [27], a heuristic for adjusting task weights based on rates of loss changes; **(5)** Uncertainty Weighting (UW) [18] uses task uncertainty as a proxy to adjust task weights; **(6)** Random Loss Weighting (RLW) [23] that samples task weighting whose log-probabilities follow the normal distribution; **(7)** MGDA [35] that finds the equal descent direction for each task; **(8)** PCGRAD [43] proposes to project each task gradient to the normal plan of that of other tasks and combining them together in the end; **(9)** CAGRAD [24] optimizes the average loss while explicitly controls the minimum decrease across tasks; **(10)** IMTL-G [25] finds the update direction with equal projections on task gradients; **(11)** GRADDROP [7] that randomly dropout certain dimensions of the task gradients based on how much they conflict; **(12)** NASHMTL [32] formulates MTL as a bargaining game and finds the solution to the game that benefits all tasks. For FAMO, we choose the best hyperparameter $\gamma \in \{0.0001, 0.001, 0.01\}$ based on the validation loss. Specifically, we choose $\gamma$ equals 0.01 for the CityScapes dataset and 0.001 for the rest of the datasets. See Appendix E for results with error bars.

**Evaluations:** We consider two metrics [32] for MTL: **1)** $\Delta m$%, the average per-task performance drop of a method $m$ relative to the STL baseline denoted as $b$: $\Delta m\% = \frac{1}{K} \sum_{k=1}^{K} (-1)^{\delta_k} (M_{m,k} - M_{b,k})/M_{b,k} \times 100$, where $M_{b,k}$ and $M_{m,k}$ are the STL and $m$'s value for metric $M_k$. $\delta_k = 1$ (or 0) if the $M_k$ is higher (or lower) the better. **2) Mean Rank (MR)**: the average rank of each method across tasks. For instance, if a method ranks first for every task, **MR** will be 1.

**Findings:** Results on the four benchmark datasets are provided in Table 1, 2 and 3. We observe that FAMO performs consistently well across different supervised learning MTL benchmarks compared

| Method | CityScapes | | | | | | CelebA | |
| | Segmentation | | Depth | | MR ↓ | $\Delta m\%$ ↓ | MR ↓ | $\Delta m\%$ ↓ |
| | mIoU ↑ | Pix Acc ↑ | Abs Err ↓ | Rel Err ↓ | | | | |
| STL | 74.01 | 93.16 | 0.0125 | 27.77 | | | | |
| LS | 70.95 | 91.73 | 0.0161 | 33.83 | 6.50 | 14.11 | 4.15 | 6.28 |
| SI | 70.95 | 91.73 | 0.0161 | 33.83 | 9.25 | 14.11 | 7.20 | 7.83 |
| RLW | 74.57 | 93.41 | 0.0158 | 47.79 | 9.25 | 24.38 | 1.46 | 5.22 |
| DWA | 75.24 | 93.52 | 0.0160 | 44.37 | 6.50 | 21.45 | 3.20 | 6.95 |
| UW | 72.02 | 92.85 | 0.0140 | **30.13** | 6.00 | **5.89** | 3.23 | 5.78 |
| MGDA | 68.84 | 91.54 | 0.0309 | 33.50 | 9.75 | 44.14 | 14.85 | 10.93 |
| PCGRAD | 75.13 | 93.48 | 0.0154 | 42.07 | 6.75 | 18.29 | 3.17 | 6.65 |
| GRADDROP | 75.27 | 93.53 | 0.0157 | 47.54 | 6.00 | 23.73 | 3.29 | 7.80 |
| CAGRAD | 75.16 | 93.48 | 0.0141 | 37.60 | 5.75 | 11.64 | 2.48 | 6.20 |
| IMTL-G | 75.33 | 93.49 | 0.0135 | 38.41 | 4.00 | 11.10 | **0.84** | **4.67** |
| NASHMTL | **75.41** | **93.66** | **0.0129** | 35.02 | **2.00** | 6.82 | 2.84 | 4.97 |
| FAMO | 74.54 | 93.29 | 0.0145 | 32.59 | 6.25 | 8.13 | 1.21 | 4.72 |

**Table 3:** Results on CityScapes (2 tasks) and CelebA (40 tasks) datasets. Each experiment is repeated over 3 random seeds and the mean is reported. The best average result is marked in bold. **MR** and $\Delta m\%$ are the main metrics for MTL performance.

to other gradient manipulation methods. In particular, it achieves state-of-the-art results in terms of $\Delta m\%$ on the NYU-v2 and QM-9 datasets.

**Multitask Reinforcement Learning.** We further apply FAMO to multitask reinforcement learning (MTRL) problems as MTRL often suffers more from conflicting gradients due to the stochastic nature of reinforcement learning [43]. Following CAGRAD [24], we apply FAMO on the MetaWorld [44] MT10 benchmark, which consists of 10 robot manipulation tasks with different reward functions. Following [37], we use Soft Actor-Critic (SAC) [15] as the underlying RL algorithm, and compare against baseline methods including LS (SAC with a shared model) [44], Soft Modularization [42] (an MTL network that routes different modules in a shared model to form different policies), PC-GRAD [43], CAGRAD and NASHMTL [32]. The experimental setting and hyperparameters all match exactly with those in CAGRAD. For NASHMTL, we report the results of applying the NASHMTL update once per $\{1, 50, 100\}$ iterations.[4] The results for all methods are provided in Table 5.2.

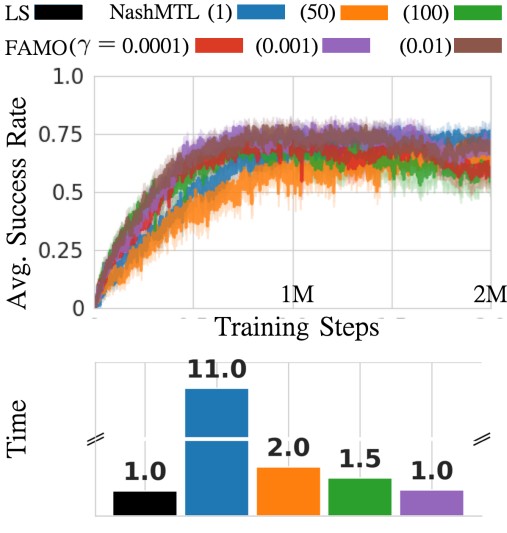

**Figure 3:** Training Success Rate and Time.

| Method | Success ↑ (mean ± stderr) |
| --- | --- |
| LS (lower bound) | 0.49 ±0.07 |
| STL (proxy for upper bound) | 0.90 ±0.03 |
| PCGRAD [43] | 0.72 ±0.02 |
| SOFT MODULARIZATION [42] | 0.73 ±0.04 |
| CAGRAD | 0.83 ±0.05 |
| NASHMTL [32] (every 1) | **0.91** ±0.03 |
| NASHMTL [32] (every 50) | 0.85 ±0.02 |
| NASHMTL [32] (every 100) | 0.87 ±0.03 |
| NASHMTL (ours) (every 1) | 0.80 ±0.13 |
| NASHMTL (ours) (every 50) | 0.76 ±0.10 |
| NASHMTL (ours) (every 100) | 0.80 ±0.12 |
| UW [18] | 0.77 ±0.05 |
| FAMO (ours) | 0.83 ±0.05 |

**Table 4:** MTRL results (averaged over 10 runs) on the Metaworld-10 benchmark.

[4]We could not reproduce the MTRL results of NASHMTL exactly, so we report both the results from the original paper and our reproduced results.

**Findings:** From Table 5.2, we observe that FAMO performs comparably to CAGRAD and outperforms PCGRAD and the average gradient descent baselines by a large margin. FAMO also outperforms NASHMTL based on our implementation. Moreover, FAMO is significantly faster than NASHMTL, even when it is applied once every 100 steps.

## 5.3 MTL Efficiency (Training Time Comparison)

Figure 4 provides the FAMO's average training time per epoch against that of the baseline methods.

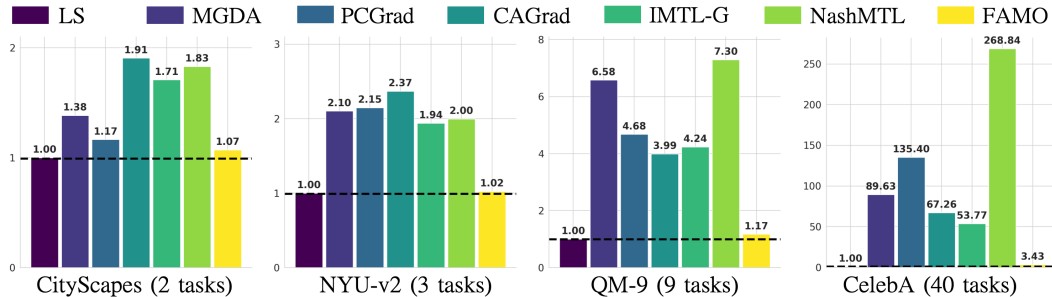

**Figure 4:** Average training time per epoch for different MTL optimization methods. We report the relative training time of a method to that of the linear scalarization (LS) method (which uses the average gradient).

**Findings:** From the figure, we observe that FAMO introduces negligible overhead across all benchmark datasets compared to the LS method, which is, in theory, the lower bound for computation time. In contrast, methods like NASHMTL have much longer training time compared to FAMO. More importantly, the computation cost of these methods scales with the number of tasks. In addition, note that these methods also take at least $\mathcal{O}(K)$ space to store the task gradients, which is implausible for large models in the many-task setting (i.e., when $m = |\theta|$ and $K$ are large).

## 5.4 Ablation on $\gamma$

In this section, we provide the ablation study on the regularization coefficient $\gamma$ in Figure 5.

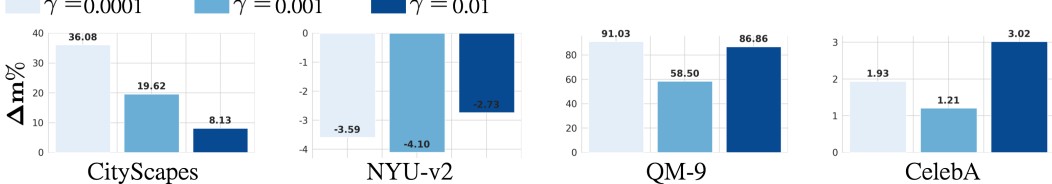

**Figure 5:** Ablation over $\gamma$: we plot the performance of FAMO (in terms of $\boldsymbol{\Delta m}\%$ using different values of $\gamma$ from $\{0.0001, 0.001, 0.01\}$ on the four supervised MTL benchmarks.

**Findings:** From Figure 5, we can observe that choosing the right regularization coefficient can be crucial. But except for CityScapes, FAMO performs reasonably well using all different $\gamma$s. The problem with CityScapes is that one of the task losses is close to 0 at the very beginning, hence small changes in task weighting can result in very different loss improvement. Therefore we conjecture that using a larger $\gamma$, in this case, can help stabilize MTL.

## 6 Conclusion and Limitations

In this work, we introduce FAMO, a fast optimization method for multitask learning (MTL) that mitigates the conflicting gradients using $\mathcal{O}(1)$ space and time. As multitasking large models gain more attention, we believe designing efficient but effective optimizers like FAMO for MTL is crucial. FAMO balances task losses by ensuring each task's loss decreases approximately at an equal rate. Empirically, we observe that FAMO can achieve competitive performance against the state-of-the-art MTL gradient manipulation methods. One limitation of FAMO is its dependency on the regularization parameter $\gamma$, which is introduced due to the stochastic update of the task weighting logits $\boldsymbol{w}$. Future work can investigate a more principled way of determining $\gamma$.

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

# A  Gradient Manipulation Methods

In this section, we provide a brief overview of representative gradient manipulation methods in multitask/multiobjective optimization. Specifically, we will also discuss the connections among these methods.

**Multiple Gradient Descent Algorithm (MGDA) [9, 35]**  The MGDA algorithm is one of the earliest gradient manipulation methods for multitask learning. In MGDA, the per step update $d_t$ is found by solving

$$\max_{d \in \mathbb{R}^m} \min_{i \in [k]} \nabla \ell_{i,t}^\top d - \frac{1}{2}\|d\|^2.$$

As a result, the solution $d^*$ of MGDA optimizes the "worst improvement" across all tasks or equivalently seeks an *equal* descent across all task losses as much as possible. But in practice, MGDA suffers from slow convergence since the update $d^*$ can be very small. For instance, if one task has a very small loss scale, the progress of all other tasks will be bounded by the progress on this task. Note that the original objective in (6) is similar to the MGDA objective in the sense that we can view optimizing (6) as optimizing the log of the task losses. Hence, when we compare FAMO against MGDA, one can regard FAMO as balancing the *rate* of loss improvement while MGDA balances the absolute improvement across task losses.

**Projecting Gradient Descent (PCGRAD) [43]**  PCGRAD initializes $v_{\text{PC}}^i = \nabla \ell_{i,t}$, then for each task $i$, PCGRAD loops over all task $j \neq i$ (in a random order, which is crucial as mentioned in [43]) and removes the "conflict"

$$v_{\text{PC}}^i \leftarrow v_{\text{PC}}^i - \frac{{v_{\text{PC}}^i}^\top \nabla \ell_{j,t}}{\|\ell_{j,t}\|^2} \nabla \ell_{j,t} \quad \text{if} \quad {v_{\text{PC}}^i}^\top \nabla \ell_{j,t} < 0.$$

In the end, PCGRAD produces $d_t = \frac{1}{k}\sum_{i=1}^k v_{\text{PC}}^i$. Due to the construction, PCGRAD will also help improve the "worst improvement" across all tasks since the "conflicts" have been removed. However, due to the stochastic iterative procedural of this algorithm, it is hard to understand PCGRAD from a first principle approach.

**Conflict-averse Gradient Descent (CAGRAD) [24]**  $d_t$ is found by solving

$$\max_{d \in \mathbb{R}^m} \min_{i \in [k]} \nabla \ell_{i,t}^\top d \quad \text{s.t.} \quad \|d - \nabla \ell_{0,t}\| \leq c \|\nabla \ell_{0,t}\|.$$

Here, $\ell_{0,t} = \frac{1}{k}\sum_{i=1}^k \ell_{i,t}$. CAGRAD seeks an update $d_t$ that optimizes the "worst improvement" as much as possible, conditioned on that the update still decreases the average loss. By controlling the hyperparameter $c$, CAGRAD can recover MGDA ($c \to \infty$) and the vanilla averaged gradient descent ($c \to 0$). Due to the extra constraint, CAGRAD provably converges to the stationary points of $\ell_0$ when $0 \leq c < 1$.

**Impartial Multi-Task Learning (IMTL-G) [25]**  IMTL-G finds $d_t$ such that it shares the same cosine similarity with any task gradients:

$$\forall i \neq j, \quad d_t^\top \frac{\nabla \ell_{i,t}}{\|\nabla \ell_{i,t}\|} = d_t^\top \frac{\nabla \ell_{j,t}}{\|\nabla \ell_{j,t}\|}, \quad \text{and} \quad d_t = \sum_{i=1}^k w_{i,t} \nabla \ell_{i,t}, \text{ for some } w_t \in \mathbb{S}_k.$$

The constraint that $d_t = \sum_{i=1}^k w_{i,t} \nabla \ell_{i,t}$ is for preventing the problem from being under-determined. From the above equation, we can see that IMTL-G ignores the "size" of each task gradient and only cares about the "direction". As a result, **one can think of IMTL-G as a variant of MGDA that applies to the normalized gradients**. By doing so, IMTL-G does not suffer from the straggler effect due to slow objectives. Furthermore, one can **view IMTL-G as the equal angle descent**, which is also proposed in Katrutsa et al. [17], where the objective is to find $d$ such that

$$\forall i \neq j, \qquad \cos(d, \nabla \ell_{i,t}) = \cos(d, \nabla \ell_{j,t}).$$

**NASHMTL[32]** NASHMTL finds $d_t$ by solving a bargaining game treating the local improvement of each task loss as the utility for each task:

$$\max_{d \in \mathbb{R}^m, \|d\| \leq 1} \sum_{i=1}^{k} \log \left( \nabla \ell_{i,t}^{\top} d \right).$$

Note that the objective of NASHMTL implicitly assumes that there exists $d$ such that $\forall\ i, \ \nabla \ell_{i,t}^{\top} d > 0$ (otherwise we reach the Pareto front). It is easy to see that

$$\max_{\|d\| \leq 1} \sum_{i=1}^{k} \log \left( \nabla \ell_{i,t}^{\top} d \right) = \max_{\|d\| \leq 1} \sum_{i=1}^{k} \log \langle \frac{\nabla \ell_{i,t}}{\|\nabla \ell_{i,t}\|}, d \rangle = \max_{\|d\| \leq 1} \sum_{i=1}^{k} \log \cos \left( \nabla \ell_{i,t}, d \right).$$

Therefore, due to the $\log$, NASHMTL also ignores the "size" of task gradients and only cares about their "directions". Moreover, denote $u_i = \frac{\nabla \ell_{i,t}}{\|\nabla \ell_{i,t}\|}$. Then, according to the KKT condition, we know:

$$\sum_i \frac{u_i}{u_i^{\top} d} - \alpha d = 0, \quad \alpha \geq 0 \qquad \Longrightarrow \qquad d = \frac{1}{\alpha} \sum_i \frac{1}{u_i^{\top} d} u_i.$$

Consider when $k = 2$, if we take the *equal angle descent* direction: $d_{\angle} = (u_1 + u_2)/2$ (note that as $u_1$ and $u_2$ are normalized, their bisector is just their average). Then it is easy to check that

$$d_{\angle} = \frac{1}{\alpha} \left( \frac{2}{u_1^{\top}(u_1 + u_2)} u_1 + \frac{2}{u_2^{\top}(u_1 + u_2)} u_2 \right), \quad \text{where} \ \alpha = \frac{u_1^{\top}(u_1 + u_2)}{4} = \frac{u_2^{\top}(u_1 + u_2)}{4}.$$

As a result, we can see that **when $k = 2$, NASHMTL is equivalent to IMTL-G (or the equal angle descent)**. However, when $k > 2$, this is not in general true.

**Remark** Note that all of these gradient manipulation methods require computing and storing $K$ task gradients before applying $f$ to compute $d_t$, which often involves solving an additional optimization problem. Hence, these methods can be slow for large $K$ and large model sizes.

## B Amortizing other Gradient Manipulation Methods

Although FAMO uses iterative update on $w$, it is not immediately clear whether we can apply the same amortization easily on other existing gradient manipulation methods. In this section, we discuss such possibilities and point out the challenges.

**Amortizing MGDA** This is almost the same as in FAMO, except that MGDA acts on the original task losses while FAMO acts on the log of task losses.

**Amortizing PCGRAD** For PCGRAD, finding the final update vector requires iteratively projecting one task gradient to the other, so there is no straightforward way of bypassing the computation of task gradients.

**Amortizing IMTL-G** The task weighting in IMTL-G is computed by a series of matrix-matrix and matrix-vector products using task gradients [25]. Hence, it is also hard to amortize its computation over time.

Therefore, we focus on deriving the amortization for CAGRAD and NASHMTL.

**Amortizing CAGRAD** For CAGRAD, the dual objective is

$$\min_{w \in \mathbb{S}_k} F(w) = g_w^{\top} g_0 + c \|g_w\| \|g_0\|, \tag{15}$$

where $g_0 = \nabla \ell_{0,t}$ and $g_w = \sum_{i=1}^{k} w_i \nabla \ell_i$. Denote

$$G = \begin{bmatrix} \nabla \ell_{1,t}^{\top} \\ \vdots \\ \nabla \ell_{k,t}^{\top} \end{bmatrix}.$$

Now, if we take the gradient with respect to $w$ in (15), we have:

$$\frac{\partial F}{\partial w} = G^\top g_0 + c\frac{\|g_0\|}{\|g_w\|}G^\top g_w. \tag{16}$$

As a result, in order to approximate this gradient, one can separately estimate:

$$
\begin{aligned}
G^\top g_0 &\approx \frac{\ell(\theta) - \ell(\theta - \alpha g_0)}{\alpha} \\
G^\top g_w &\approx \frac{\ell(\theta) - \ell(\theta - \alpha g_w)}{\alpha} \\
\|g_0\| &\approx \sqrt{1^\top G^\top g_0} \\
\|g_w\| &\approx \sqrt{w^\top G^\top g_w}
\end{aligned}
\tag{17}
$$

Once all these are estimated, one can combine them together to perform a single update on $w$. But note that this will require 3 forward and backward passes through the model, making it harder to implement in practice.

**Amortizing NASHMTL**     Per derivation from NASHMTL [32], the objective is to solve for $w$:

$$G^\top Gw = 1 \oslash w. \tag{18}$$

One can therefore form an objective:

$$\min_w F(w) = \left\| G^\top Gw - 1 \oslash w \right\|_2^2. \tag{19}$$

Taking the derivative of $F$ with respect to $w$, we have

$$\frac{\partial F}{\partial w} = 2G^\top G\left(G^\top g_w - 1 \oslash w\right) + 2\left(G^\top g_w - 1 \oslash w\right) \oslash (w \odot w). \tag{20}$$

Therefore, to approximate the gradient of $w$, one needs to first estimate

$$G^\top g_w \approx \frac{L(\theta) - L(\theta - \alpha g_w)}{\alpha} = \eta. \tag{21}$$

Then we estimate

$$G^\top G(\eta - 1 \oslash w) \approx \frac{L(\theta) - L(\theta - \alpha G(\eta - 1 \oslash w))}{\alpha}. \tag{22}$$

Again, this results in 3 forward and backward passes through the model, let alone the overhead of resetting the model back to $\theta$ (requires a copy of the original weights).

In short, though it is possible to derive fast approximation algorithm to approximate the gradient update on $w$ for some of the existing gradient manipulation methods, it often involves much more complicated computation compared to that of FAMO.

## C   FAMO Pseudocode in PyTorch

We provide the pseudocode for FAMO in Algorithm 2. To use FAMO, one just first compute the task losses, call `get_weighted_loss` to get the weighted loss, and do the normal backpropagation through the weighted loss. After that, one call `update` to update the task weighting.

## D   Toy Example

We provide the task objectives for the toy example in the following. The model parameter $\theta = (\theta_1, \theta_2) \in \mathbb{R}^2$ and the task objectives are $L^1$ and $L^2$:

$$L^1(\theta) = 0.1 \cdot (c_1(\theta)f_1(\theta) + c_2(\theta)g_1(\theta)) \ \text{ and } \ L^2(\theta) = c_1(\theta)f_2(\theta) + c_2(\theta)g_2(\theta), \ \text{where}$$

$$f_1(\theta) = \log\left(\max(|0.5(-\theta_1 - 7) - \tanh(-\theta_2)|, \ 0.000005)\right) + 6,$$

$$f_2(\theta) = \log\left(\max(|0.5(-\theta_1 + 3) - \tanh(-\theta_2) + 2|, \ 0.000005)\right) + 6,$$

$$g_1(\theta) = \left((-\theta_1 + 7)^2 + 0.1 * (-\theta_2 - 8)^2\right)/10 - 20,$$

$$g_2(\theta) = \left((-\theta_1 - 7)^2 + 0.1 * (-\theta_2 - 8)^2\right)/10 - 20,$$

$$c_1(\theta) = \max(\tanh(0.5 * \theta_2), \ 0) \ \text{ and } \ c_2(\theta) = \max(\tanh(-0.5 * \theta_2), \ 0).$$

**Algorithm 2** Implementation of FAMO in PyTorch-like Pseudocode

```python
class FAMO:
def __init__(self, num_tasks, min_losses, α=0.025, γ=0.001):
    # min_losses  (num_tasks,) the loss lower bound for each task.
    self.min_losses = min_losses
    self.xi = torch.tensor([0.0] * num_tasks, requires_grad=True)
    self.xi_opt = torch.optim.Adam([self.xi], lr=α, weight_decay=γ)

def get_weighted_loss(self, losses):
    # losses  (num_tasks,)
    z = F.softmax(self.xi, -1)
    D = losses - self.min_losses + 1e-8
    c = 1 / (z / D).sum().detach()
    loss = (c * D.log() * z).sum()
    return loss

def update(self, prev_losses, curr_losses):
    # prev_losses  (num_tasks,)
    # curr_losses  (num_tasks,)
    delta = (prev_losses - self.min_losses + 1e-8).log() -
            (curr_losses - self.min_losses + 1e-8).log()
    with torch.enable_grad():
        d = torch.autograd.grad(F.softmax(self.xi, -1),
                                self.xi,
                                grad_outputs=delta.detach())[0]
    self.xi_opt.zero_grad()
    self.xi.grad = d
    self.xi_opt.step
```

# E   Experimental Results with Error Bars

We followed the exact experimental setup from NASHMTL [32]. Therefore, the numbers for baseline methods are taken from their original paper. In the following, we provide FAMO's result with error bars.

| Method | Segmentation | | Depth | | Surface Normal | | | | | $\mathbf{\Delta}m\% \downarrow$ |
|---|---|---|---|---|---|---|---|---|---|---|
| | | | | | Angle Dist $\downarrow$ | | Within $t° \uparrow$ | | | |
| | mIoU $\uparrow$ | Pix Acc $\uparrow$ | Abs Err $\downarrow$ | Rel Err $\downarrow$ | Mean | Median | 11.25 | 22.5 | 30 | |
| FAMO (mean) | 38.88 | 64.90 | 0.5474 | 0.2194 | 25.06 | 19.57 | 29.21 | 56.61 | 68.98 | -4.10 |
| FAMO (stderr) | ±0.54 | ±0.21 | ±0.0016 | ±0.0026 | ±0.06 | ±0.09 | ±0.17 | ±0.19 | ±0.14 | ±0.39 |

**Table 5:** Results on NYU-v2 dataset (3 tasks). Each experiment is repeated over 3 random seeds and the mean is reported. The best average result is marked in bold. **MR** and $\mathbf{\Delta}m\%$ are the main metrics for MTL performance.

| Method | $\mu$ | $\alpha$ | $\epsilon_{\text{HOMO}}$ | $\epsilon_{\text{LUMO}}$ | $\langle R^2 \rangle$ | ZPVE | $U_0$ | $U$ | $H$ | $G$ | $c_v$ | $\mathbf{\Delta}m\% \downarrow$ |
|---|---|---|---|---|---|---|---|---|---|---|---|---|
| | | | | | | MAE $\downarrow$ | | | | | | |
| FAMO (mean) | 0.15 | 0.30 | 94.0 | 95.2 | 1.63 | 4.95 | 70.82 | 71.2 | 71.2 | 70.3 | 0.10 | 58.5 |
| FAMO (stderr) | ±0.0046 | ±0.0070 | ±3.074 | ±2.413 | ±0.0211 | ±0.0871 | ±2.17 | ±2.19 | ±2.19 | ±2.21 | ±0.0026 | ±3.26 |

**Table 6:** Results on QM-9 dataset (11 tasks). Each experiment is repeated over 3 random seeds and the mean is reported. The best average result is marked in bold. **MR** and $\mathbf{\Delta}m\%$ are the main metrics for MTL performance.

| Method | CityScapes | | | | | CelebA |
|---|---|---|---|---|---|---|
| | Segmentation | | Depth | | $\mathbf{\Delta}m\% \downarrow$ | $\mathbf{\Delta}m\% \downarrow$ |
| | mIoU $\uparrow$ | Pix Acc $\uparrow$ | Abs Err $\downarrow$ | Rel Err $\downarrow$ | | |
| FAMO (mean) | 74.54 | 93.29 | 0.0145 | 32.59 | 8.13 | 1.21 |
| FAMO (stderr) | ±0.11 | ±0.04 | ±0.0009 | ±1.06 | ±1.98 | ±0.24 |

**Table 7:** Results on CityScapes (2 tasks) and CelebA (40 tasks) datasets. Each experiment is repeated over 3 random seeds and the mean is reported. The best average result is marked in bold. **MR** and $\mathbf{\Delta}m\%$ are the main metrics for MTL performance.

