# OpenReview forum: "FAMO: Fast Adaptive Multitask Optimization"
_NeurIPS.cc/2023/Conference — NeurIPS 2023 poster_

### Official Review · Reviewer_otYb · 2023-06-11

**Soundness:** 3 good
**Presentation:** 3 good
**Contribution:** 3 good
**Rating:** 7
**Confidence:** 4

**Summary:**

The paper addresses an important issue with (most) prior MTL optimization techniques which requires the computation of all per-task gradients, during training, for obtaining the update direction. This results in a $\mathcal{O}(K)$ requirement in space and time where $K$ is the number of tasks. The authors propose a novel approach for dynamically weighting tasks using $\mathcal{O}(1)$ space and time.

**Strengths:**

1. The paper proposes a novel approach for addressing a known pain point with (most) SoTA multi-task optimization methods.
2. The paper provides strong empirical evidence which indicates the method performs on par with SoTA MTL methods (e.g., NashMTL, IMTL-G, CAGrad) while significantly reducing the optimization time.
3. The paper is well-written and well-structured. Connection to prior works in MTL literature is well established (Appendix A-B).


**Weaknesses:**

1. While the division of the proposed method follows a different trajectory, the proposed algorithm is fairly similar to MGDA. Hence, more discussion is needed on the connection between the proposed method and MGDA.
2. Please provide additional details on the experimental sections (data split sizes, HPs, etc.).
3. Hyperparameter (HP) optimization:
    - Previous papers on MTL (e.g., CAGrad) did not include a validation split for NYU/Cityscapes. If the authors follow the same settings, how did you tune FAMO’s specific HPs?
    - Missing details on the tuning of the lr for the task weights.
4. From section 5.4 (and as stated by the authors), it appears the method is highly sensitive to the choice of HPs.
5. It would be beneficial to include figures of the losses throughout the optimization process and some metric to verify that the losses decrease in an (approximately) equal rate.
6. It would be beneficial if you could provide a figure of the task weights throughout the optimization process.

Minor:
Table 2, MR column missing indication for best-performing method.


**Questions:**

1. Could you provide some intuition as to why the proposed method alleviates the CG problem in MTL, only by ensuring each task’s loss decreases at an equal rate?
2. Have you tried performing several optimization steps for approximating the solution in Eq. 7 at each training iteration?


**Limitations:**

Limitations addressed.

---

> ### Author Rebuttal · Authors · 2023-08-08
>
> We thank the reviewer for the constructive feedback and comments. We address your concerns and questions in the following.
>
> ---
>
> **Weakness:**
>
> **1. More discussion is needed with MGDA.**
>
> Thanks for the suggestion, we will make the difference and connection to MGDA more clear in the final version of the paper. Note that we have provided some comparisons in Appendix A already.
>
> **2. Add additional experimental details.**
>
> Thanks for the suggestion and we will add more details in the appendix. Moreover, we will release all code for reproducing all experiments, to make it easy for others to build on our research.
>
> **3.. Hyperparameters optimization.**
>
> We follow the same criterion as in CAGrad: we use the training loss to determine the hyperparameter.
>
> **4. Sensitivity to hyperparameters.**
>
> We admit that to achieve the best performance of FAMO currently it requires some tuning of the $\gamma$ parameter. But FAMO's performance remains strong across a relatively wide range of $\gamma$s. Moreover, empirically we observe that setting it between $0.001$ to $0.01$ achieves good MTL performance. As stated in the conclusion and limitation section (Sec. 6), an interesting future direction is to make FAMO robust.
>
> **5. It will be good to include figures showing that losses are decreasing at roughly equal rates.**
>
> Thanks for the suggestion, we provide it in Figure 2 above. We are showing the descent rate over the epoch for both FAMO and NashMTL. We observe that the rates for different losses are more aligned in FAMO than in NashMTL.
>
> **6. Provide a task weight throughout the optimization process.**
>
> Thanks for the suggestion, we have provided a task weighting plot in Figure 3 above. From the plot, we see that both FAMO and NashMTL essentially gradually increase the task weight on task 3 (the surface normal prediction), which is the task that has the minimum loss scale.
>
> ---
>
> **Questions:**
>
> **1. More intuitions on why FAMO mitigates CG.**
>
> Mitigating CG is one objective, but the method should also optimize the task losses. Regarding why FAMO mitigates CG, it is because CG happens when a certain task is under-optimized due to an imbalance weighting among tasks. FAMO’s objective is to encourage an equal descent rate for all task objectives. Hence if CG happens, then definitely this objective is violated as well. Furthermore, while MGDA also mitigates CG, it encourages equal descent, which may not make sense when one of the task losses is extremely small (hence all other tasks' progress is bounded by this single task). Therefore, we believe FAMO’s objective is more reasonable in practice.
>
> **2. Have you tried multiple steps for approximating Eq.7?**
>
> Thanks for the advice. We have not yet tried multiple steps for approximating Eq.7. In principle it is doable. But performing multiple steps approximation will no longer only require 1 extra forward pass. It will introduce more overhead and slow down FAMO. It is an interesting future direction to investigate the trade-off between speed and performance.

---

> > ### Comment · Reviewer_otYb · 2023-08-16
> >
> > Thanks for the response and for providing additional results.
> >
> > > Hyperparameters optimization.
> >
> > Using the training loss appears to me like a bad practice. For example, we can increase the network's capacity and achieves lower training loss, but this definitely not implies lower generalization error. However, if you are using the same procedure for all methods, I assume it is reasonable (in terms of fairness).
> >
> > > FAMO mitigates CG.
> >
> > I do not agree with the statement "CG happens when a certain task is under-optimized due to an imbalance weighting among tasks". It is not clear without additional theoretical or empirical evidence.
> >
> > In general, I'm fairly happy with the authors' response and I will like to maintain my initial score.

---

### Official Review · Reviewer_DG6u · 2023-06-27

**Soundness:** 2 fair
**Presentation:** 2 fair
**Contribution:** 2 fair
**Rating:** 6
**Confidence:** 4

**Summary:**

This paper proposes a new multi-task balancing method with claimed O(1) efficiency, which is much more efficient than previous gradient manipulation methods.

The core idea is to let the multiple losses decrease at roughly the same speed.



**Strengths:**

The method is more efficient than previous gradient manipulation methods.
Authors conducted lots of experiments on several datasets and show that the method is performing well on them.

**Weaknesses:**

1. I think the contribution is little bit over-stated. When I read the title, abstract and introduction, I thought authors will present a novel method to manipulate task gradients with O(1) time and space complexity, which is awesome. Why I thought that? For example, in Figure 1, they compare FAMO with other gradient balancing methods like PCGrad and CAGrad, which balance task gradients with O(K) time and space complexity.

However, authors actually do not manipulate gradients but weight losses. Losses weighting are not novel path with previous methods DWA and UW. If you balance losses rather gradients, of course you can have O(1) time and space complexity since you do not touch parameter gradients w.r.t to K losses.
In Question [highlight in the introduction], authors want to design an MTL optimizer, which should handle the gradients of parameters. I am not sure if the algorithm proposed in the paper can be called a new optimizer?
My suggestion is that authors can re-factory the story [title, abs, intro] as an efficient loss balancing methods, which seems more suitable to their contributions.

2.  The method part is not presented very clearly, see the questions.

**Questions:**

1. Equation 6 is transformed from Equation 5 using L'Hopital's rule, based on the step size \alpha is reaching to 0, however, learning rate \alpha is normally something like 0.001, which is much larger than 0. So, Equation 6 is an approximation here and should be made clear.

2. Could authors elaborate a bit on why Equation 7 is the dual form for (6)?

3. In Section 3.2,"But then, assume [author's equation but hard to write for me here], one can approximate (8) by:", may I ask why we can assume this equation? what is the intuition for this assumption?

4. What is w? the final weighs losses I guess?. In the paper, authors have used v_{t} and z_{t} as task weight. And they did not clearly explain w.


**Limitations:**

No social impact

---

> ### Author Rebuttal · Authors · 2023-08-08
>
> We thank the reviewer for the constructive feedback and comments. We address your concerns and questions in the following.
>
> ---
>
> **Weakness:**
>
> **1. FAMO does not manipulate gradients but reweighs them. Re-weighting methods exist like UW/DWA. Should refactor the abstract, intro, method.**
>
> Existing gradient manipulation methods all produce the final manipulated gradient as a linear combination of task gradients. Hence from this perspective, they are also re-weighting methods. We think the confusion originates from the definition of “manipulation”. In our case, we mean that the final update is some function of the task gradients for achieving a certain property. From this perspective, UW or DWA approach the MTL problems from the “loss” perspective, i.e., they propose a way to manipulate losses. While gradient manipulation methods will not necessarily propose a meaningful new objective to optimize, they aim to manipulate the task gradients such that after performing the update, certain a property is achieved. In our case, it is that the task losses descend at roughly an equal rate. We will make the above more clear in the revised version of the paper.
>
> Apart from the above, we point out that we compare FAMO against UW and DWA in our tables. We emphasize that re-weighting methods, so far, do not reach similar performance compared to state-of-the-art gradient manipulation methods.
>
> **2. Methods are not clear**
>
> We address your concerns in the following.
>
> ---
>
> **Questions:**
>
> **1. Eq.6 is an approximation of Eq.5, which should be made clear.**
>
> We will make sure it is clear in the final revised version of the paper.
>
> **2. Why Eq.7 is the dual form of Eq.6?**
>
> The proof is shown on page 16 in the appendix. In summary, the objective is convex and we can hence switch the min and max.
>
> **3. What is the intuition in Sec. 3.2 from Eq.8 to Eq.9?**
>
> The transition from Eq. 8 to Eq. 9 follows the first-order Taylor expansion. Specifically, denote $h(\theta_t) = \log \Delta_t$,
>
> $$
> h(\theta_t) - h(\theta_{t+1}) = h(\theta_t) - h(\theta_t - \eta \nabla \log \Delta_t z) \approx \eta \bigg(\nabla_\theta h_t\bigg)^\top \bigg( \nabla \log \Delta_t z\bigg) = \eta  (\nabla \log \Delta_t) ^\top (\nabla \log \Delta_t) z.
> $$
>
> We will make it clear in the final version of the paper.
>
> **4. What is $w$, the final weight loss?**
>
> $w$ is the logits to the task weights (check Alg. 1). For instance, $z = \textbf{softmax}(w)$.

---

> > ### Comment · Reviewer_DG6u · 2023-08-15
> > **Questions to Rebuttal by Authors**
> >
> > Thanks for Authors' reply!
> > I feel my original rating is not high enough.
> >
> > Just a few more questions to confirm my understanding:
> > (1) In equation 10, the gradient of w is basically the gradient of z multiply with the derivative of softmax(w), right? where is 1/eta in equation 9?
> >
> > (2) So this is the first time of using amortized computation in gradient manipulation methods? how did the previous methods like MGDA, CAGrad and IMTL-G to solve z, directly solve z at every iteration using some convex optimizer like Frank-Wolfe?
> >
> > (3) In the new experiments Figure1, seems amortized MGDA is the worst, so this means the trade-off between efficiency and performance?
> > Why amortized MGDA is so bad?

---

> > > ### Author Response · Authors · 2023-08-15
> > > **Response to The Reviewer**
> > >
> > > We sincerely thank the reviewer for acknowledging the contribution of FAMO. Regarding your questions:
> > >
> > > **1. In Eq. 10, the gradient of $w$ is the gradient of $z$ multiplied by the derivative of softmax, where is $1/\eta$?**
> > >
> > > Yes, you are correct, Eq. 10 is just the direct application of the chain rule. The $1/\eta$ is subsumed into $\beta$.
> > >
> > > **2. So this is the first time using amortized computation in gradient manipulation methods? how did the previous methods like MGDA, CAGrad, and IMTL-G solve z, directly solve z at every iteration using some convex optimizer like Frank-Wolfe?**
> > >
> > > To our knowledge, we are the first to derive such an amortized computation based on the nature of the algorithm. Previous methods like MGDA, CAGrad, or IMTL-G explicitly solve an inner optimization problem to compute the new update direction. And you are right, such a problem could be solved by Frank-Wolfe style methods [1].
> > >
> > > **3. Why amortized MGDA is so bad?**
> > >
> > > We believe the reason is due to the fact that different task objectives have different loss scales and MGDA is agnostic to the loss scale and only seeks the best "equal descent" direction. Therefore, as the third task loss is very small, the progress on the first 2 tasks is limited by that. As a thought experiment, imagine the model parameter is initialized near a local optimum for one of the task objectives, then MGDA will make very little progress (so as amortized MGDA). By contrast, FAMO considers the "relative progress" instead of the "absolute progress".
> > >
> > > ---
> > >
> > > **Reference:**
> > >
> > > [1] Multi-Task Learning as Multi-Objective Optimization.

---

> > > > ### Comment · Reviewer_DG6u · 2023-08-16
> > > >
> > > > Thanks for the reply, I have raised my rating from 4 to 6

---

### Official Review · Reviewer_XmqU · 2023-07-06

**Soundness:** 3 good
**Presentation:** 3 good
**Contribution:** 2 fair
**Rating:** 5
**Confidence:** 4

**Summary:**

This paper proposes a dynamic convex combination multi-task learning losses reweighting so that it can decrease different task losses more balancedly while having little computational overhead than simple task loss average. Specifically, they formulate an optimization problem to find a new “gradient” direction that can achieve sufficient relative-to-magnitude decrease for the worst task loss. Instead of solving this optimization exactly every update, the authors amortize the optimization by keeping dual variables on the probability simplex across updates and update them through one-step gradient descent after every primal variable (learnable parameter) update. Experimentally, the authors show that their proposed method FAMO can achieve comparable/superior on 4 multitask supervised learning benchmarks and 1 multi-task RL benchmark, while being more time-saving compared to other gradient manipulation methods.

**Strengths:**

1. The paper is in general well-written, with a nice discussion about different multi-task gradient manipulation technique in Appendix A.
2. The proposed method FAMO empirically is competitive with other methods on multiple benchmarks.

**Weaknesses:**


**[Lack of proof for significant contribution]**

In terms of methodology, as the authors mention themselves, FAMO’s motivation formulation is basically applying MGDA to the logarithm of all task losses. Thus the formulation is practical but not very novel. What distinguishes FAMO from MGDA fundamentally is FAMO’s amortized optimization of the dual variable. I will raise additional question later in this section about solidifying this contribution. In terms of experiment results, the authors claim FAMO can perform better/equal to other baseline methods in terms of (1) final performance, and (2) wall-clock time. For final performance, it is not convincing enough that FAMO can always match/outperform other MTL methods. On CityScapes and CelebA, it seems to fall behind some other methods. On QM-9, it’s also not clear to me what’s the value of multi-task learning — it seems single task learning (STL) is better (with lower MAE values) than almost all MTL methods  (including FAMO) (please do correct me if I’m misunderstanding this). I would appreciate authors to provide further clarifications about these points. In terms of wall-clock time, the authors only show the wall clock time for each epoch for different methods. However, this neglects the fact that different methods might require different number of epochs to converge. It is conceivable that a non-optimization-amortized method could take longer per epoch but converge in fewer epochs than FAMO and thus could use less total time. I think the authors should also report the total time as it should be final metric to judge the wall-clock savings.

**[Understanding the impact of amortizing the optimization]** A primary contribution of the paper is to propose to amortize the optimization to find the update direction instead of solving it exactly each update. Hence I believe it is important and useful that the authors provide experiment results on the non-amortized version of FAMO, where the optimization in equation (5) is solved exactly in each iteration. With this non-amortized result, it would provide the context of whether the amortization deteriorates the performance, allowing the readers to understand the room of improvement with amortization. Besides, it would also be interesting to see the amortized vs non-amortized comparison for MGDA to further understand this point.

**[Potentially incorrect proof for proposition 3.1]** In the proof for proposition 3.1, the authors use the Lagrangian of the dual by adding $cz^\top \mathbf{1}$ to the objective to get rid of the simplex constraint. However, the simplex constraint is not only about sum of the variables being 1, but also about having nonnegative individual values. Thus, I believe these nonnegative constraint should also be reflected in the Lagrangian. Otherwise it needs to be specifically argued that the optimal solution cannot occur on the edges of the simplex.

**Questions:**

**[Why is FAMO $O(1)$ in space?]** For performing backprop over the simple case of the average task loss (LS), either all the $K$ tasks’ computation graphs are computed simultaneously and stored in the memory (needed for backprop) at the same time, or the tasks’ losses are computed one after another. In the first case, the computation graph would already have space complexity of O(K), while in the second case, the time to iterate over all tasks would have time complexity of O(K). FAMO is basically calling backprop on a reweighting of the logarithm of the task losses so it should at least have the same space and time complexity as LS. So why do the authors claim FAMO is $O(1)$ in space? Besides, storing the K weights are already technically O(K) in space. I can understand that FAMO would take less time than the other multi task gradient manipulation methods which require non-amortized optimization, but I would appreciate the authors be more rigorous about their claims with more explanation.

**[Why is STL so good?]** On QM-9 and MTRL results, it seems STL performs (much) better than other methods. Can the authors provide explainations on why that is the case?

**[Mean Rank]** In Table 3, why is IMTL-G’s mean rank (0.84) smaller than 1? On page 7 the paper says for a method that ranks first for every task, its MR is still 1.

**[Obeying the simplex constraint]** This question is relatively minor but I would suggest the authors consider alternative ways to  obey the simplex constraint other than using a softmax which couldn’t really exactly represent the corners of the simplex. Instead, one could apply $L^2$-projected gradient descent instead of gradient descent on $z$ by projecting the updated $z$ back onto the simplex. There exist efficient algorithms for simplex projection which don’t require solving the projection optimization problem iteratively: https://arxiv.org/pdf/1309.1541.pdf.

**Limitations:**

The authors mention their approach depending an extra weight decay hyperparameter as an limitation of their work. I don’t believe this paper needs to discuss negative societal impacts. I currently rate the paper as borderline accept because of the weaknesses described above. However, I’m open to increase my score given satisfactory author rebuttal responses.

---

> ### Author Rebuttal · Authors · 2023-08-08
>
> We thank the reviewer for the constructive feedback and comments. We address your concerns and questions in the following.
>
> ---
>
> **Weakness:**
>
> **1. Lack of significant contribution. Why STL is good?**
>
> The 2 contributions of FAMO are: 1) we propose that we should do equal-rate descent instead of equal descent, and 2) we introduce a fast amortizing trick that can be applied to drastically speed up the optimization, which so far only MGDA and FAMO could use with one extra forward pass in stochastic gradient descent setting.
> The STL learning performance is computed using the same network but only trained on a single task. So if we have K tasks, we essentially use K more parameters. We provide the plot of the delta M over epoch for FAMO, NashMTL, amortized MGDA, and log-MGDA (the exact algorithm of FAMO) in Figure 1. From the plot, we observe that FAMO not only is fast and performs well asymptotically, but also achieves such good performance fast over epoch.
>
> **2. Non-amortized log-MGDA.**
>
> Please refer to Figure 1 for the log-MGDA performance. It performs well but FAMO tracks its performance well.
>
> **3. Incorrect Proof**
>
> Thanks for catching that. The proof is correct with the assumption that the optimal $z$ is an interior point of $\mathcal{S}^K$, hence $z > 0$. Otherwise, we will have the extra Lagrangian multiplier $\nu \in \mathbb{R}_+^K$ such that $\nu \odot z = 0$, i.e. $\nu \odot z$ is element-wise zero. Under this case, it means that it is not possible to achieve equal rate descent with any convex combination of the gradients, the best we can hope for is that we concentrate the weighting only on a single (or a few) task gradient(s). Consider an obtuse triangle with points A, B, and C, where A-B represents $g^1$ and A-C represents $g^2$ ($g^1$, $g^2$ represent $\nabla \log \Delta^1$ and $\nabla \log \Delta^2$), and assume the angle $\angle ABC > 90\degree$, then we cannot find any update along B-C that results in equal descent. The best we can hope for is update A-B, which is $g^1$. However, not that the optimal solution is still the best possible update in terms of balancing the loss decrease rate.
>
> Note that this is the same result in the original MGDA paper, except that they are working on the original objectives while FAMO works on the log objectives. We will make sure to make it clear about our assumption in the final updated version.
>
> ---
>
> **Questions:**
>
> **1. Why is FAMO $\mathcal{O}(1)$ in space?**
>
> FAMO remains  $\mathcal{O}(1)$ in space, for the same reason that averaged gradient descent is  $\mathcal{O}(1)$ in space. This is due to the nature of the backpropagation algorithm, as explained in Line 32-34. In general, if we weigh the loss first, say $L_w = w_1 L^1 + w_2 L^2$, since we know this weighting $w_1$ and $w_2$ beforehand, one can take advantage of the backpropagation as $$w_1 \frac{d L^1}{d \theta} + w_2 \frac{d L^2}{d \theta} = \frac{d L_w}{d \theta},$$ which can be computed within 1 backward pass **without** extra space. This is similar to the fact that if we compute the loss over a batch, the space to store the gradient will not scale with the batch size. This is because we average (or sum) the batch loss first before doing the backpropagation, and taking the derivative is a linear operation.
>
> **2. Why is STL so good?**
>
> It is addressed in response to weakness 1. We mention it here again for convenience. STL reported in this paper means we use the same neural architecture but only train for a single task. So if we have $K$ tasks it means we need to train $K$ models separately. The total number of effective parameters is much larger than that of an MTL model.
>
> **3. Mean Rank < 1**
>
> Thanks for catching that. It is a typo. The Mean Rank for Celeb-A should all be increased by 1. This is because in our implementation we first compute the Mean Rank (0-indexed) and then add 1 on top of it. But due to a mistake, we forgot the add 1 for the ranks in Celeb-A. It will be corrected in the final version of the paper.
>
> **4. Alternative to reparametrization (projected gradient)**
>
> Thanks for the suggestion. We agree that it would be interesting to investigate whether performing projected gradient descent on w can be more efficient for FAMO. We adopt the reparameterization because then the problem becomes completely unconstrained. Usually constrained optimization is harder to solve stably in practice.

---

> > ### Comment · Reviewer_XmqU · 2023-08-12
> > **Follow-up to authors' rebuttal (Part I)**
> >
> > I want to thank the reviewer’s rebuttal response and the effort in creating new figures to compare FAMO with non-amortized log-MGDA. I have the following comments:
> >
> > 1. **Understanding the two contributions separately**
> >
> > As also pointed out by Reviewer gjX4, the two contributions of the paper **[** **1)** using logarithm of task losses in MGDA) and **2)** using amortized optimization for fast optimization **]** are two separate ideas and deserve analyses separately. Previously, the main paper hasn’t analyzed these two points separately, but directly provide the results of the combination of these two ideas through FAMO. In the rebuttal, the authors have already made an improvement in providing the result of log-MGDA on the NYU-v2 dataset in Figure 1. This allows us to understand 1) and 2) separately. To further strengthen these analyses, I believe it is important to add the result of log-MGDA on every dataset in Section 5.2, and in addition, also through the time measurements in Section 5.3. This will give the readers the complete information about the delta of improvement in each of the two ideas separately. With these changes, the authors might also need to add additional sentences in **Findings** to describe these results.
> >
> > 2. **Proposition 3.1 is indeed incorrect**
> >
> > As the authors have agreed, without an additional assumption that the optimal $z$ is an interior point of $\mathcal{S}^k$, the conclusion of proposition 3.1 does not hold. However, I don’t think the authors should fix the proof by simply adding this assumption because this assumption is too restrictive: for example, when $g^1 \coloneqq \nabla \log \Delta^1 $ and $g^2 \coloneqq \nabla \log \Delta^2$ are pointing in exactly the same direction (thus no conflicting gradient), as long as $g^2$ is longer than $g^1$, the norm minimizing solution in the 1-simplex should put all of its weight on $g^1$ and none of its weight on $g^2$. In this case, the optimal $z$ **has to be on the corner (instead of the interior)** of the 1-simplex and there is **NO equal rate** of improvement across these two task losses. Based on this simple example, the authors should not claim FAMO (or log-MGDA) to be aiming at equal rate of task loss decrease (this is mentioned in line 41 as the first contribution).
> >
> > The authors claim that proposition 3.1 is the same result as that in the original paper except that the authors are working with log objectives. However, the original MGDA paper never claims they are optimizing for equal **amount** of decrease across task losses. Instead, the original MGDA paper only aims to find a solution on the pareto-frontier by considering multi-task learning as a multi-objective optimization problem. Thus the authors also shouldn’t claim log-MGDA or FAMO as optimizing for equal **rate** of decrease across task losses.
> >
> > In terms of high-level intuition, the authors’ original formulation of the objective in equation (5) is a min-max problem, where the goal is *not to ensure a notion of equality* among the tasks but only to improve the rate of improvement of the *worst performing task*. This intuition would also imply that getting equal rate of improvement to be unlikely under this min-max formulation.

---

> > > ### Author Response · Authors · 2023-08-12
> > > **Response to the Reviewer's Response**
> > >
> > > We thank the reviewer for the prompt reply and address your further comments in the following.
> > >
> > > **1. Understanding the two contributions separately.**
> > >
> > > Thanks for your suggestion and we will make sure the log-MGDA results for every experiment are added in the final version of the paper.
> > >
> > > **2. Proposition 3.1 is indeed incorrect.**
> > >
> > > We thank the reviewer for catching this. Essentially Proposition 3.1 will be similar to Theorem 2.2 (ii) in the MGDA paper [1] (which also states that MGDA can ensure an equal descent when the found update direction is an interior point). We will ensure the message of Proposition 3.1 is accurate and correct in the final updated version. We will remove the claim that FAMO always ensures an equal rate. But we want to emphasize (as the reviewer also pointed out) that the mini-max objective is seeking the **best possible worst-case** loss decrease rate. From this perspective, FAMO always seeks the best possible equal rate descent direction within the convex combination of all task gradients. We will make sure this point is clear in our final updated version of the paper.
> > >
> > > **3. Why is FAMO $\mathcal{O}(1)$ in space?**
> > >
> > > We believe the confusion results from the difference between multitask and multiobjective learning. In many of the MTL benchmarks (like the ones we use in this work and in previous literature), it is in fact multiobjective, meaning that we use the same data point, pass it through the same architecture, and then form $K$ losses to optimize. In this case, FAMO is strictly better than most existing MTL methods that require computing task gradients individually, both in space and time. In the more general multitask learning setting, where we have different data for each task, then the computation graph is indeed $\mathcal{O}(K)$ for both FAMO and LS (this is similar to the mini-batch gradient descent case where the computation graph is $K$ times larger due to the batch dimension). But the computation time is still $\mathcal{O}(1)$ for FAMO and LS. We will make this point clear in the updated version of the paper.
> > >
> > > **4. Further Clarifying Goals of Multitask Learning.**
> > >
> > > We thank the reviewer for the suggestion and agree that the current MTL research community lacks larger and more interesting benchmarks where we can see better generalization when learning more tasks simultaneously. In addition, we believe that eventually when we go to thousands or even millions of tasks, it is almost impossible to have a single model for each task. So though the MTL method might still underperform the single-task model, it is expected as the single-task model essentially has $K$ times more parameters, but we cannot afford that many parameters during training. Therefore, we believe as we enter an even larger data regime, then the core research problem in MTL is to push forward the boundary of MTL performance using a single model.
> > >
> > > **Reference:**
> > >
> > > [1] Multiple-gradient descent algorithm (MGDA) for multiobjective optimization.

---

> > > > ### Comment · Reviewer_XmqU · 2023-08-16
> > > > **Reviewer's summary**
> > > >
> > > > I would like to thank the authors for their prompt responses to my follow-up questions. I appreciate the authors’ clarification about the point of FAMO being $O(1)$ in space and time and I was indeed *misunderstanding* the fact that the benchmarks the authors used are multi-objective (instead of input-set-disjoint multiple tasks) learning. This also to a large extent _resolves my confusion_ about the fact that having one model for each learning objective (STL) can outperform having a single model for multiple objectives (MTL).
> > > >
> > > > In summary of the discussions between the authors and myself, I’m generally happy with the authors’ responses, which have answered many of my questions. The only two things I have reservations for are things that I can only feel confident about after seeing the actual revised version of the paper:  1) The authors promise to include the result of the non-amortized FAMO result (i.e. log-MGDA) on the rest of the experiments and also provide relevant discussions about these new results. (This is important to separately understand each of the two ideas the authors proposed.) 2) The authors acknowledge to change the claims about FAMO (including but not limited to Proposition 3.1) so that it’s not advertised as an equal rate method for all task without qualifications. Given these two points, I’ll keep my current score but I hope that my reviews is of use to the authors to further improve their paper. I also want to thank the authors for their efforts in the paper and rebuttal.

---

> > ### Comment · Reviewer_XmqU · 2023-08-12
> > **Follow-up to authors' rebuttal (Part II)**
> >
> > 3. **Why is FAMO $O(1)$ in space?**
> >
> > I acknowledge the authors’ intension to claim that FAMO only needs to store $1$ model gradients but not $K$ gradients. However, what I originally want to point out is that claiming FAMO (or averaged loss gradient descent) is $O(1)$ in space is a bit ambiguous. This is because the space requirement in reverse mode (backprop) gradient computation does **not only** depend on the size of parameters whose gradient is to be computed, but **more importantly** also on the entire computation graph these parameters are involved in. (This is because we need to store all the intermediate computations in memory for the backward gradient computation). In the case of FAMO and averaged loss gradient descent, a single model’s parameters are still involved in the computation of $K$ different losses. Thus the entire computation graph’s size would still scale with $O(K)$ instead of $O(1)$. I would recommend the authors make a distinction between the space used to compute the gradients (which unavoidably is $O(K)$ for all methods) and the space needed to store the necessary gradients and use it to compute the final update direction (FAMO is $O(1)$ in this regard).
> >
> > 4. **Further clarifying goals of multitask learning**
> >
> > I believe the typical narrative of multitask learning is to improve the performance of each task by leveraging the similarity across tasks. However, many experiments in the paper (QM-9, CityScapes, CelebA, and MRTL) all show that doing multitask learning cannot outperform single task learning in terms of performance. In contrast, as the authors have explained in the rebuttal response, the benefit is instead in model storage, where we sacrifice a bit of individual task performance (compared to STL) in order to avoid storing $O(K)$ models but only $1$ model. Maybe this is an implicit assumption common to researchers working with these benchmarks, but I would recommend the authors be more explicit about this in the main paper.

---

### Official Review · Reviewer_gjX4 · 2023-07-12

**Soundness:** 3 good
**Presentation:** 4 excellent
**Contribution:** 3 good
**Rating:** 6
**Confidence:** 4

**Summary:**

This paper proposed a novel multi-task optimization method aimed at mitigating conflict between task gradients without inducing the substantial time slowdown that typically comes with specialized multi-task optimizers.
The proposed method, called FAMO, aims at improving the worst-case rate of improvement across all tasks at each step.
At a high level, it does this by setting the weights of each task to minimize the gradient norm of the log task loss (similar to how MGDA sets weights by minimizing the gradient norm of the task loss). They show that finding these weights is the dual of finding the direction that maximizes the minimum rate of improvement across tasks. In practice however, these weights are learned over time via SGD updates after each optimization step of the main model, rather than optimized in full at each model update step.

The authors then demonstrate their method on a toy 2D loss landscape (used in prior work), showing that their method reaches the Pareto frontier of this problem, like other optimization methods (and unliked naive Adam), but has the closest overall compute time to Adam. Next, the authors compare their method to 11 other optimization methods for MTL on 4 supervised MTL and 1 RL setting finding that FAMO compares competitively (it is consistently in the top 2 or 3 performing methods) in all settings.  Finally, the authors demonstrate the performance efficiency of their method in the supervised learning settings showing that FAMO is slightly faster than comparable methods on settings with 2 or 3 tasks, and orders of magnitude faster as the number of tasks increases to 10 and 60.

**Strengths:**

- The method being proposed has clear benefits in terms of speed / efficiency, especially as the number of tasks scales, which is very important in many MTL settings.
- The method is compared to a broad and representative set of optimization methods, and performs competitively on all benchmarks proposed even outperforming slower methods consistently.
- Moreover, the experimental methodology seems sound; all experiments consider multiple random seeds and 2 sensible metrics to compare methods are considered.
- The paper is well written, with good figures, clear descriptions, and the motivation is straightforward. Moreover, I find the proposed method to be explained well.

**Weaknesses:**

- The primary contribution of FAMO (in my view) is the decision to maximize minimum task rate of improvement, by focusing on the gradient of the log loss, rather than maximizing the minimum total improvement (i.e. MGDA). However, this is not the “fast” part. The fast part of the method comes from optimizing the task-weights slowly, via gradient updates, throughout the optimization process. The authors do well to point this out in the appendix, but the effects of these contributions should be analyzed separately. In a deep neural network it is possible that the generalization benefits come from the amortized optimization of task weights, rather than the specific weight objectives. This could easily be addressed by, for instance, comparing amortized MGDA to FAMO, which is perhaps the most comparable setting.
- I feel as though GradNorm is also natural comparison to this method, given that it also learns task weights during training (in an amortized fashion), with the goal of balancing the average rate of improvement across tasks.
- [21] is referenced as suggesting that current optimization methods aren’t worth the tradeoff in computation time, but iirc their conclusion was that optimization methods actually don’t help performance over the baseline, which appears to not be true in this papers experiments. This discrepancy should likely be commented on.

**Questions:**

- Is there a typo in the psuedo code of algo 2? z is computed via softmax, but then the raw weights w are used for the rest of the get_weighted_loss method.

**Limitations:**

yes

---

> ### Author Rebuttal · Authors · 2023-08-08
>
> We thank the reviewer for the constructive feedback and comments. We address your concerns and questions in the following.
>
> ---
>
> **Weakness:**
>
> **1. Compare FAMO against amortized MGDA.**
>
> Please refer to Figure 1 above. We see that since amortized MGDA (like MGDA) is seeking equal descent for each loss, it performs poorly compared to FAMO.
>
> **2. Compare FAMO against GradNorm.**
>
> Please refer to Figure 2 in [1] for a comparison of IMTL-G versus GradNorm on ResNet-50. It is known that GradNorm does not perform as well as gradient manipulation methods. Moreover, GradNorm actually is $\mathcal{O}(K)$ in space and time, because in order to perform the GradNorm update, one needs to take the gradient of each task’s gradient norm. Hence it requires computing each task’s gradient first.
>
> **3. [21] claims that MTL methods are not useful.**
> - [21] only studies MGDA, PCGrad, RLW, and GradDrop. It does not consider more recent gradient manipulation methods like CAGrad, NashMTL, IMTL-G, etc.
> - For supervised learning, [21] only tries multi-MNIST (too simple) and Cityscapes (only 2-task). It hasn’t been tried on celeb or NYU-v2.
> - We understand that what [21] claims is that there always exists a set of unit scalarizations that performs on par with existing gradient manipulation methods. But it does not provide a method for how we can decide this set of task weights except for just doing a grid-search, and this scales poorly with the number of tasks. On the other hand, the main drawback of existing gradient manipulation methods is that they are too slow to be applied. Hence we find that FAMO achieves the sweet spot in that it achieves the state-of-the-art performance across a wide range of commonly used MTL benchmarks and it is very cheap to apply in practice.
>
> ---
>
> **Questions:**
>
> **1. Typo in Pseudocode.**
>
> Thanks for catching that! We will correct it in the final version.
>
> ---
>
> **Reference:**
>
>
> [1] [Towards Impartial Multi-task Learning](https://openreview.net/pdf?id=IMPnRXEWpvr).

---

### Author Rebuttal · Authors · 2023-08-08

## Common Response to All Reviewers (with Additional Results)

---

We sincerely thank all reviewers for their comments and valuable suggestions. Per the reviewers' request, **we conduct additional experiments on NYU-v2 and summarize the results in the attached PDF. We will respond to each reviewer's questions and concerns separately via individual comments.**

---

**A. Additional Results**

**1.** MTL performance of FAMO, compared against **log-MGDA** (the exact form of FAMO), **amortized MGDA** (using FAMO's idea but applied to MGDA), over **epochs** of training (Figure 1).

**Findings:**
- Amortized MGDA performs poorly, as it pursues equal descent instead of equal rate descent.
- log-MGDA performs well and outperforms Nash-MTL at the end of training. But FAMO tracks log-MGDA's performance closely while being much faster (log-MGDA has the same computation efficiency as MGDA).
- FAMO not only is an efficient algorithm that achieves good asymptotic performance but also achieves good performance fast over epochs of training.

**2.** We provide the loss decrease rate $\frac{L^i_t - L^i_{t+1}}{L^i_{t}}$ for LS, NashMTL, and FAMO over epochs (Figure 2).

**Findings:**
- Among the three methods, we only observe that FAMO's losses decrease at a roughly equal rate. The decrease rate in NashMTL is also roughly even but not as good as in FAMO. LS has the most imbalanced loss decrease rate.
- Note that there is a bump in the middle of training caused by the learning rate scheduling (i.e., we halved the learning rate at the 100-th epoch)

**3.** We provide the task weights for the three task objectives over training steps for LS, NashMTL and FAMO (Figure 3).

**Findings:**
- NashMTL, due to its formulation, has task weights not necessarily sum to 1.
- Both NashMTL and FAMO gradually assign a higher task weighting to Task 3 (surface normal prediction), which is the task that has the minimum scale. By contrast, LS always assigns equal task weight to all three tasks.

---

**B. FAMO's contribution**

Some reviewers have concerns about FAMO's contribution. We want to emphasize 2 contributions of FAMO:
- We propose that the equal rate objective is more reasonable than the equal descent objective. And we empirically verify this claim.
- We identify a simple and efficient way of amortizing the log-MGDA. So far, only MGDA and log-MGDA, among existing MTL gradient manipulation methods, can be easily amortized by having a single extra forward pass of the network.

Combining the two contributions, the introduced FAMO algorithm achieves state-of-the-art MTL performance across popular MTL benchmarks, while having almost no overhead compared to the vanilla learning scalarization (LS) method. This will be particularly important if we go to hundreds or even a larger number of tasks, where existing gradient manipulation methods would almost certainly fail to apply due to their expensive computation.

The implementation of FAMO, together with all code to reproduce all experiment results, will be made available public upon acceptance.

---

### Decision · Program_Chairs · 2023-09-21

**Decision:**

Accept (poster)

**Comment:**

This paper presents Fast Adaptive Multitask Optimization (FAMO), an optimizer for multi-task learning problems that aims to mitigate the conflicting gradient problem with decreased space and time requirements by iteratively updating task weightings while ensuring that all task losses decrease at roughly similar rates. The reviewers appreciated the clear empirical benefits of the proposed approach, as validated through extensive experimental evaluation. However, they also raised several substantive issues in their initial reviews, including:
* lack of assessment of the impact of the paper's two contributions separately (maximizing the minimum task rate of improvement vs. optimizing the task weights slowly via gradient updates). The proposed fix is to compare to log-MGDA in all experiments, and provide relevant discussions about the results.
* incorrectness of the proof of Proposition 3.1, notably since FAMO only achieves equal rate of task loss decrease under particular restrictive assumptions. The proposed fix is to adapt the message of Proposition 3.1 to clarify this, and remove the claim that FAMO always ensures an equal rate (rather than the best possible worst-case loss decrease rate).
* need for better discussion of the relationship between FAMO and MGDA. The proposed fix is to add more discussion of this in the paper.

The reviewers were satisfied by the authors' clarification on these concerns, under the assumption that the authors will make the above changes to the paper in the final version.